

# $^{13}$C based proteinogenic amino acid (PAA) and metabolic flux ratio analysis of *Lactococcus lactis* reveals changes in pentose phosphate (PP) pathway in response to agitation and temperature related stresses

Kamalrul Azlan Azizan[1,*], Habtom W. Ressom[2,*], Eduardo R. Mendoza[3,4,*] and Syarul Nataqain Baharum[1,*]

[1] Metabolomics Research Laboratory, Institute of Systems Biology (INBIOSIS), Universiti Kebangsaan Malaysia (UKM), Bangi, Selangor, Malaysia

[2] Departments of Oncology, Georgetown Lombardi Comprehensive Cancer Center, Georgetown University Medical Center, Washington, D.C., United States of America

[3] Institute of Mathematics, University of the Philippines Diliman, Quezon City, Metro Manila, Philippines

[4] Membrane Biochemistry Group, Max Planck Institute of Biochemistry, Planegg, Germany

[*] These authors contributed equally to this work.

Corresponding author
Syarul Nataqain Baharum,
nataqain@ukm.edu.my

## ABSTRACT

*Lactococcus lactis* subsp. *cremoris* MG1363 is an important starter culture for dairy fermentation. During industrial fermentations, *L. lactis* is constantly exposed to stresses that affect the growth and performance of the bacterium. Although the response of *L. lactis* to several stresses has been described, the adaptation mechanisms at the level of *in vivo* fluxes have seldom been described. To gain insights into cellular metabolism, $^{13}$C metabolic flux analysis and gas chromatography mass spectrometry (GC-MS) were used to measure the flux ratios of active pathways in the central metabolism of *L. lactis* when subjected to three conditions varying in temperature (30 °C, 37 °C) and agitation (with and without agitation at 150 rpm). Collectively, the concentrations of proteinogenic amino acids (PAAs) and free fatty acids (FAAs) were compared, and Pearson correlation analysis ($r$) was calculated to measure the pairwise relationship between PAAs. Branched chain and aromatic amino acids, threonine, serine, lysine and histidine were correlated strongly, suggesting changes in flux regulation in glycolysis, the pentose phosphate (PP) pathway, malic enzyme and anaplerotic reaction catalysed by pyruvate carboxylase (pycA). Flux ratio analysis revealed that glucose was mainly converted by glycolysis, highlighting the stability of *L. lactis*' central carbon metabolism despite different conditions. Higher flux ratios through oxaloacetate (OAA) from pyruvate (PYR) reaction in all conditions suggested the activation of pyruvate carboxylate (pycA) in *L. lactis*, in response to acid stress during exponential phase. Subsequently, more significant flux ratio differences were seen through the oxidative and non-oxidative pentose phosphate (PP) pathways, malic enzyme, and serine and C1 metabolism, suggesting NADPH requirements in response to environmental stimuli. These reactions could play an important role in optimization strategies for metabolic engineering in *L. lactis*. Overall, the integration of systematic analysis of amino acids
and flux ratio analysis provides a systems-level understanding of how *L. lactis* regulates central metabolism under various conditions.

# INTRODUCTION

*Lactococcus lactis* is a lactic acid bacterium (LAB) that produces lactate as its main catabolic by-product (*Gaspar et al., 2013*; *Zhao et al., 2013*). The bacterium is generally recognized as safe (GRAS) and is the primary constituent in many artisanal starter cultures (*Brandsma et al., 2012*; *Taibi et al., 2011*). In addition to producing lactate, *L. lactis* is also known to produce organoleptic properties that contribute to flavor, taste and texture (*Azizan, Baharum & Mohd Noor, 2012*; *Dhaisne et al., 2013*). However, the flavor formation ability of *L. lactis* depends on the generation and uptake of amino acids by the bacterium (*Garcia-Cayuela et al., 2012*; *Tanous et al., 2005*; *Van Kranenburg et al., 2002*). Amino acids are essential for the growth of *L. lactis* (*Adamberg et al., 2009*; *Ayad et al., 1999*; *Lahtvee et al., 2011*). It was demonstrated that *L. lactis* is auxotrophic for several amino acids that they cannot synthesize from simpler nitrogen sources (*Ayad et al., 1999*; *Trip, Mulder & Lolkema, 2013*; *Van Kranenburg et al., 2002*; *Wegmann et al., 2007*). Amino acids are important in biochemical reactions and metabolic networks (*Cocuron, Tsogtbaatar & Alonso, 2017*). They are involved as building blocks for protein synthesis, as signaling molecules and also in determining physiological conditions (*Tanaka et al., 2013*).

Previously, the comprehensive metabolome and transcriptome profiles of *L. lactis* under different conditions have been described (*Chen et al., 2013*; *Dijkstra et al., 2014a*; *Dijkstra et al., 2014b*; *Ibrahim et al., 2010*; *Papagianni & Avramidis, 2011*; *Taibi et al., 2011*). Despite extensive progress, however, the adaptation mechanisms of *L. lactis* to specific stresses have rarely been investigated at the level of *in vivo* fractional fluxes or flux ratios. Particularly, it is important to gain insight into the metabolic network of *L. lactis* for better manipulation of *L. lactis* as a cell factory. Understanding metabolic fluxes is a prerequisite for fluxomics by providing insights into cellular physiology and changes in the activity of intracellular pathways. The mechanisms underlying metabolic fluxes are of great interest because they can provide better interpretation of an observed phenotype (*Weindl et al., 2016*). Subsequently, the control of metabolic fluxes allows better design of metabolic systems (*Sriyudthsak, Shiraishi & Hirai, 2013*).

The [13]C isotopic labeling approach is the most powerful method for metabolic flux determination in biological systems. The approach is mostly applied in microbes to probe their central carbon metabolism (*Antoniewicz, 2013*). Often proteinogenic amino acids (PAAs) are used to trace [13]C patterns as they are related to the precursor molecules that are key components of central metabolism (*Zamboni et al., 2009*). Metabolic flux ratio analysis is an alternative approach used to calculate the *in vivo* flux activity and to investigate intermediary central carbon metabolism (*Tao et al., 2012*). The approach

uses simple network topology, [13]C-enrichment patterns obtained using NMR or MS and predefined analytic formulas to measure the relative flux contributions of interesting metabolic reactions (*Fischer & Sauer, 2003*; *Xiong et al., 2010*). Meanwhile, the use of pattern recognition methods such as principal component analysis (PCA) and correlation analysis ($r$) to identify phenotypic variation of metabolite labeling patterns has been recently reported (*Chua et al., 2013*; *Tanaka et al., 2013*). Correlation analysis in particular allows classification of groups of samples according to their common and/or unique signatures. The analysis is useful to uncover specific physiologic conditions or states via the correlation patterns.

In this study, we aim to compare the central metabolism of *L. lactis* when subjected to three conditions varying in temperature (30 °C, 37 °C) and agitation (with and without agitation 150 rpm). Using [13]C labeled glucose of different compositions and gas chromatography-mass spectrometry (GC-MS), we examined changes in the proteinogenic amino acids (PAAs) and measured the fractional distribution of active pathways in the central metabolism. Briefly, *L. lactis* subsp. *cremoris* MG1363 was cultivated in batch cultures at 30 °C with (30WA) and without agitation (30WOA) and at 37 °C without agitation (37WOA). Minimal glucose (0.5 g/l) with different compositions of [13]C-labelled and unlabelled glucose was used as sole carbon source. Flux ratios of active pathways were then calculated and compared between the different conditions. Collectively, the concentrations of proteinogenic amino acids (PAAs) and free fatty acids (FAAs) obtained from intra- and extracellular were compared. Subsequently, Pearson correlation ($r$) analysis was carried out to investigate the pairwise relationships between the PAAs, in response to different compositions of [13]C labeled glucose and different conditions. By using this approach, common amino acids that are present in all conditions were determined. Overall, the integration between systematic analysis of amino acids and fractional flux ratios provides insights into the cellular physiology state of *L. lactis* at the systems level and specifically highlights the changes in the central carbon metabolism of *L. lactis* under dynamic environmental conditions.

## MATERIAL AND METHODS

### Strains, media and growth conditions

The strain *L. lactis* subsp. *cremoris* MG1363 was obtained from Raha Abdul Rahim (Universiti Putra Malaysia). All cultivations were performed as batch cultivations with a working volume of 100 ml. Cells were grown in M17 (Oxoid Limited, Hampshire, UK) media with glucose (0.5 g/l) as carbon source. The medium was composed of ascorbic acid (0.5 g/L), $MgSO_4$ (0.25 g/L), disodium glycerophosphate (19 g/L), tryptone (5 g/L), soytone (5 g/L), beef extract (2.5 g/L) and yeast extract (2.5 g/L). A minimum of five biological replicates were prepared and incubated at 30 °C with and without agitation and at 37 °C without agitation. Cell growth was monitored using a spectrophotometer (Beckman DU, 800) by measuring optical density at 600 nm ($OD_{600}$). Glucose with an isotope label was purchased from Cambridge Isotope Laboratories, Inc. The glucose solutions were separately sterile filtered into the media. Two different compositions of glucose, a mixture

of 20% (wt/wt) [U-$^{13}$C] glucose and 80% (wt/wt) [1-$^{13}$C] glucose, and a mixture of 20% (wt/wt) [U-$^{13}$C] glucose and 80% (wt/wt) of unlabeled glucose were used. A growth curve was conducted until 8 hours to determine early, mid and late exponential phases. For cellular dry weight (CDW) determination, 10 mL of fermentation broth was centrifuged for 10 min at 4 °C at 3,000 rpm, washed with water, and dried at 90 °C for 24 h to a constant weight. Glucose and fermentation by-products (lactate, acetate, ethanol) were qualitatively measured using GC-MS. Maximum specific growth rate, specific glucose consumption were determined by regression analysis during exponential growth phase in the batch cultivation. Meanwhile, the determination of extracellular lactate, ethanol and acetate were carried out using a 600 Clarus Perkin Elmer GC-MS equipped with a Perkin Elmer Elite-WAX Polyethylene Glycol (PEG) capillary column (30-m length, 0.25-mm inside diameter and 0.50 μm thickness).

## Extraction of proteinogenic amino acid (PAA) and GC-MS analysis

A comparative analysis between different sampling times (3, 5, 6 h) of proteinogenic amino acids (PAAs) was performed to determine a suitable sampling time point for harvesting. Briefly, cell pellets were harvested at three, five, and six hours by centrifugation at 2,000 g and 4 °C. The cell pellets were then washed twice with 1 mL 0.9% NaCI and re-suspended in 200 μL of 6 M HCI at 105 °C for 16 h. The hydrolyzed cell pellets were then dried overnight on a heating block at 95 °C. For derivatization, the sample was dissolved in 20 μL of *N, N*-dimethylformamide (DMF). Derivatization was then performed using 20 μL of *N*-Methyl-*N*-ter-butyldimethylsilyl trifluoroacetamide (TBDMSTFA), and incubated in a heat block at 85 °C for 1 h. Prior to that, 50 μL of D$_4$-alanine (20 mg/L) was spiked as an internal standard. The derivatized sample was then immediately analyzed by GC-MS. 1 μL of derivatized sample was injected into a 600 Clarus Perkin Elmer GC-MS equipped with a 5MS column (30-m length, 0.25-mm inside diameter and 0.25 μm thickness; Perkin Elmer). Helium gas was used as a carrier at a flow of 1.5 mL/min. The interface and ion source temperatures were set at 250 °C and 200 °C, respectively. The electron impact voltage was set to 70 eV. The column temperature was initially set at 160 °C for 2 min, then increased to 310 °C at a rate of 20 °C/min, and then held for 0.5 min. Four fragment ions, namely [M-57]$^+$, [M-85]$^+$, [M-159]$^+$ and [f302]$^+$ of *tert*-butyldimethylsilylated amino acids, were monitored. The GC-MS data were corrected for natural abundances of O, N, H, Si and C as implemented in FiatFlux (*Nanchen, Fuhrer & Sauer, 2007*). PAAs were then validated and quantified. Concentrations were calculated using a standard curve, built using pure standard if available. Subsequently, the concentrations were corrected against D$_4$-Alanine values (internal standard) for normalization purposes.

## Extraction of free amino acid (FAA) and GC-MS analysis

Intra- and extracellular amino acids were extracted and analyzed by GC-MS as described elsewhere with modification (*Smart et al., 2010*). In short, an approximately 10 mL of filtered extracellular sample was taken and freeze dried prior to derivatization using TBDMSTFA. Meanwhile, quenching and metabolite extraction of intracellular were performed using cold ethanol and methanol:water (2:1). Each intracellular extract was then dried and derivatized using TBDMSTFA.

### $^{13}$C Metabolic flux ratio analysis using Fiat Flux

Flux ratios were calculated using RATIO module of FiatFlux (*Nanchen, Fuhrer & Sauer, 2007*; *Zamboni et al., 2009*). Briefly, the mass isotopomer distribution vectors (MDVs) of PAAs were determined from the respective mass spectra. Correction for natural isotopes of O, N, H, Si, S and C atoms in the derivatization agent and in the PAAs was carried out as implemented in FiatFlux (*Nanchen, Fuhrer & Sauer, 2007*). Using the MDVs of the PAAs, the MDVs of the respective precursor intermediates namely erythrose-4-P (E4P), phosphoenolpyruvate (PEP), pyruvate (PYR), pentose-5-P (P5P), acetyl-CoA (AcCoA), oxaloacetate (OAA), and 2-oxoglutarate (OGA) were derived. The intermediate metabolite MDVs were then used to calculate the fractional contributions of active pathways. Specifically, flux ratios were calculated for reactions in glycolysis, oxidative and non-oxidative pentose phosphate (PP) pathway, malic enzyme and anaplerotic reaction that create oxaloacetate (OAA) from pyruvate (PYR). Entner-Doudoroff (ED) pathway and glyoxylate shunt were omitted because the reactions are missing in the genome of *L. lactis* (*Flahaut et al., 2013*; *Hoefnagel et al., 2002*; *Levering et al., 2012*; *Voit, Neves & Santos, 2006a*; *Voit et al., 2006b*).

### Multivariate statistical analysis

Following data processing, a data matrix consisting of peak areas for the PAAs was generated. One-way ANOVA was performed using the web-based platform MetaboAnalyst 2.0 (*Xia et al., 2012*; *Xia et al., 2009*) to test the significance of differences in the amino acids among the culture conditions at nominal $p < 0.05$. Pearson correlation analysis was then used to explore the associations between amino acid pairs and examine the changes in the levels of proteinogenic amino acids (PAAs) in response to the different compositions of $^{13}$C labeled and unlabeled glucose and different conditions.

## RESULTS AND DISCUSSION

### Growth rate of *L. lactis* under different conditions

$^{13}$C-metabolic flux ratio analysis uses direct interpretation of $^{13}$C patterns to estimate fluxes. Therefore, the approach requires constant intracellular flux distribution during the labelling experiment. In this study we used batch cultivation to achieve constant flux distribution (steady-state). The approach enables parallel fermentation under different conditions. Initially, pre-cultivations were performed to determine the growth phases and the physiological parameters for *L. lactis when grown at* 30 °C with (30WA) and without agitation (30WOA) and at 37 °C without agitation (37WOA). The growth of the bacterium was monitored using plate counts (cfu/mL) and optical density of 600 (OD$_{600}$). The results shown in Fig. 1 indicate the growth of *L. lactis* was slow in the first three hours, before rapidly increasing after five hours and entered stationary phase after seven hours of cultivation. The total cell counts at five hours shown in Table 1 indicate high enumeration is achieved at 30WA.

The presence of lactate, acetate and ethanol were monitored via pH changes and qualitatively measured (Fig. 2). In this study, lactate was found in all conditions, suggesting a consistent activity of lactate dehydrogenase (LDH). On the other hand, the reaction is

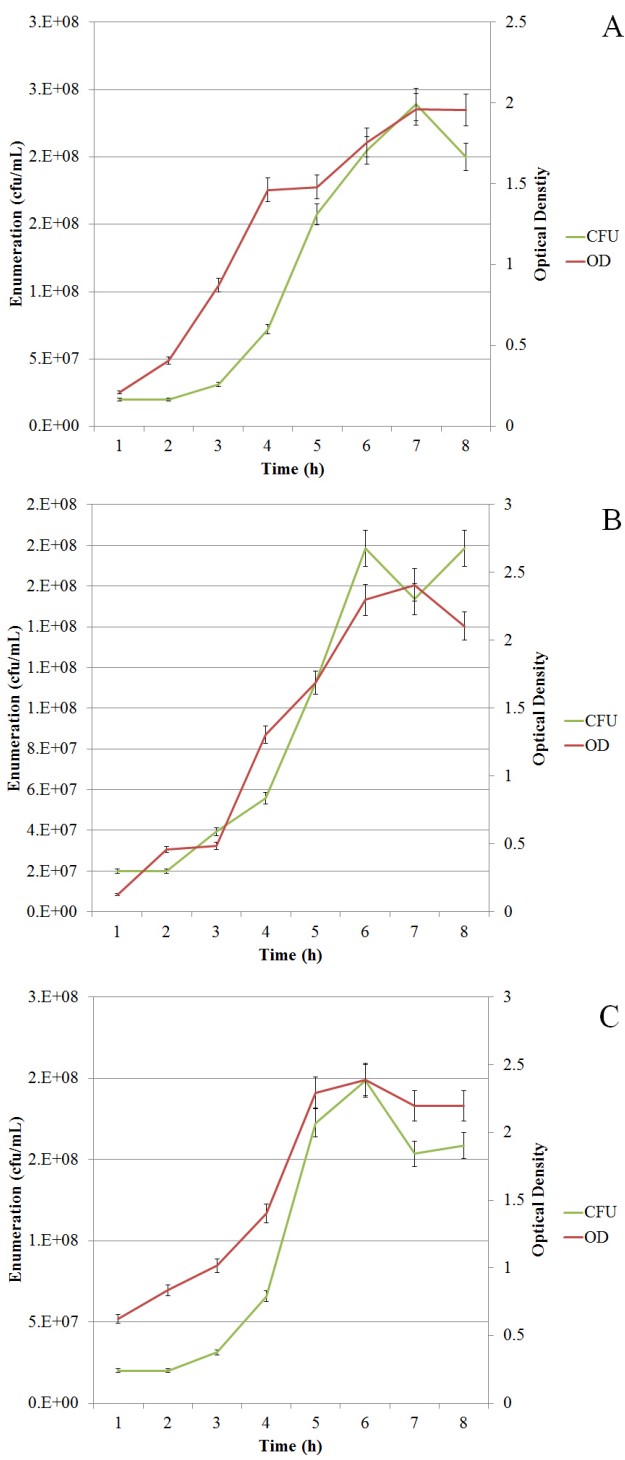

**Figure 1 Growth curves of *L. lactis*.** *L. lactis* was grown at 30 °C with agitation (150 rpm) (A), 30 °C without agitation (B), and 37 °C without agitation (C). Growth was monitored using plate counts (cfu/mL) and optical density (OD$_{600}$). The sampling point for harvesting cell pellets was determined after five hours of cultivation (mid-exponential phase). Values are given as the mean ± standard deviation ($n = 5$, biological replicates).

**Table 1** **Cell counts and pH of *L. lactis* under different conditions.** Cultivation was carried out at 30 °C with agitation (150 rpm) (30WA), 30 °C without agitation (30WOA) and 37 °C without agitation (37WOA). Measurement was carried after five hours of cultivation (mid-exponential phase). Values are given as mean ± standard deviation ($n = 5$, biological replicates).

| Parameter | 30 °C without agitation (30WOA) | 37 °C without agitation (37WOA) | 30 °C with agitation (30WA) |
|---|---|---|---|
| Cell counts | $1.13 \pm 0.6 \times 10^8$ | $1.73 \pm 0.1 \times 10^8$ | $2.08 \pm 0.4 \times 10^8$ |
| pH | $6.81 \pm 0.10$ | $6.90 \pm 0.04$ | $6.82 \pm 0.01$ |

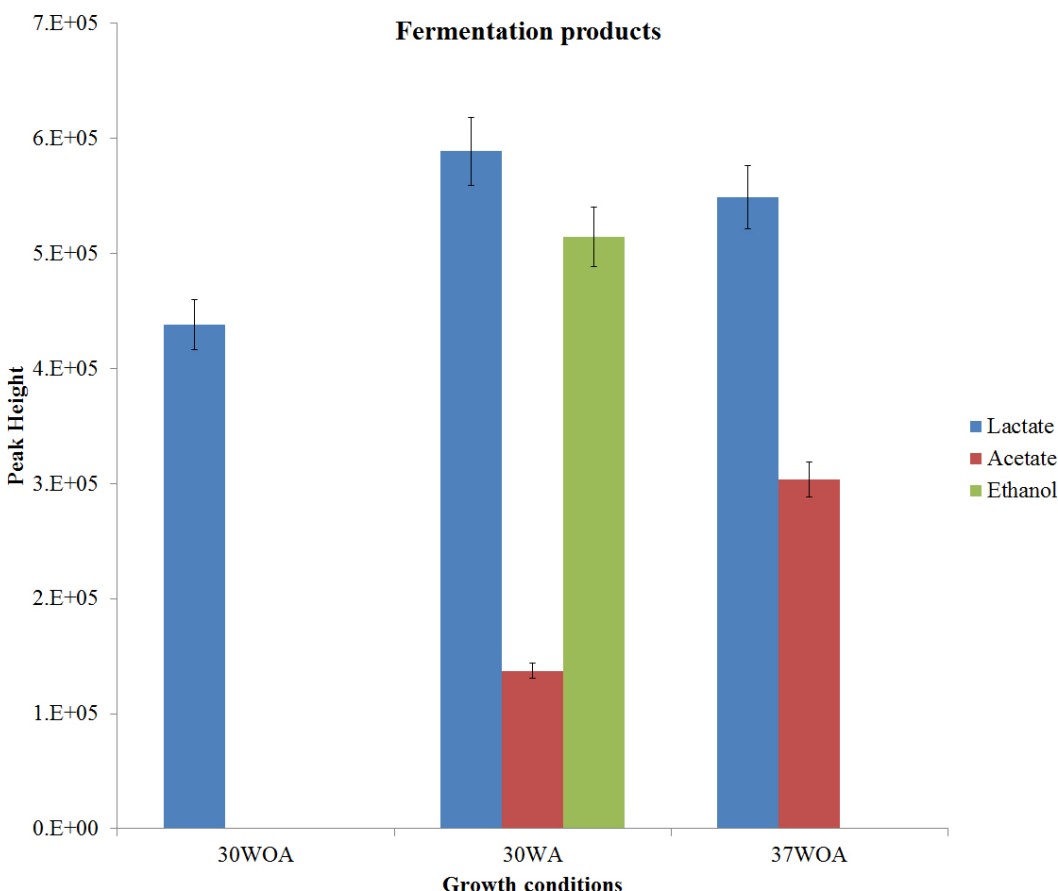

**Figure 2** **Measurement of extracellular by-products.** Measurement was obtained from 30 °C with agitation (150 rpm) (30WA), 30 °C without agitation (30WOA) and 37 °C without agitation (37WOA). Measurement was carried after five hours of cultivation (mid-exponential phase). Values are given as mean ± standard deviation ($n = 5$, biological replicates).

responsible for balancing the redox potential by oxidation of NADH to NAD$^+$ (*Feldman-Salit et al., 2013*). Meanwhile it was reported that lactate and acidification rate increased under aeration and high temperature conditions (*Chen et al., 2015*). However, in this study, no difference in acidification rate was observed (Table 1). This may due to the buffering capacity of disodium glycerophosphate to maintain the pH culture above 5.7 (*Terzaghi & Sandine, 1975*). Low pH causes inhibition of various amino acid and oligopeptide transport systems, thus affecting the growth of *L. lactis* (*Taibi et al., 2011*). Therefore *L. lactis* requires well buffered medium so that the culture pH can be maintained between 6.0 and 6.9

during growth. Notably, acetate was also detected in 30WA and 37WOA, whereas ethanol was found only in 30WA. The observation suggests a shift from homolactic to mixed-acid fermentation by *L. lactis* when exposed to agitation (30WA) and high temperature (37WOA). Additionally, a shift from formate and ethanol to acetate under glucose limited and aeration conditions has been reported previously (*Nordkvist, Jensen & Villadsen, 2003*).

Variations in the fermentation conditions are known to be contributed by several factors such as sugar consumption rate, fluxes through glycolysis, NADH/NAD$^+$ ratios, the influence of allosteric effectors on the LDH and pyruvate formate lyase (PFL) enzymes, growth conditions and effects of dissolved oxygen (*Melchiorsen et al., 2000*; *Puri et al., 2014*). Specifically, the conversion of acetate and ethanol can be initiated by pyruvate formate lyase (PFL) or pyruvate dehydrogenase (PDH), which converts PYR into acetyl-coenzyme (AcCoA). AcCoA is then converted into acetate by acetate kinase (ACK) which produces one ATP. Alternatively, AcCoA can also be reduced into ethanol by alcohol dehydrogenase (ADH), which generates NAD$^+$ from NADH.

The shift between acetate and ethanol at 30WA and 37WOA could be contributed by the oxidative stress during aeration and high temperature. It was demonstrated that aeration during *L. lactis* fermentation led to heat stress related protection, whereas a high fermentation temperature trigged robustness towards oxidative stress (*Chen et al., 2013*; *Dijkstra et al., 2014a*; *Dijkstra et al., 2014b*). In the presence of oxygen, the activity of pyruvate dehydrogenase (PDH) is reduced and could subsequently limit the supply of AcCoA and thereby limit the growth of *L. lactis*. In order to overcome this, acetate is required to compensate AcCoA (*Chen et al., 2013*). As shown in Fig. 2, accumulation of acetate and ethanol indicate the activity of PDH. The activity of NADH oxidase (NOX) may also contribute to the production of acetate and ethanol. The activity of NOX is known to be increased under aerobic conditions and play an important role in oxygen consumption, thus alleviating the effects of oxidative stress towards *L. lactis*. Moreover, the activity of NOX under aerobic conditions allows the redirection of flux from ethanol towards acetate synthesis, generating NAD$^+$ and higher yield of biomass (*Jensen et al., 2001*).

In the case of ethanol, it was reported that an increase in aeration reduced the activity of ADH and subsequently reduced the accumulation of ethanol (*Jensen et al., 2001*). However, in this study, ethanol was higher under 30WA. Meanwhile, it was reported that a small amount of ethanol was obtained when *L. lactis* was grown under microaerobic conditions but with higher growth rate (*Jensen et al., 2001*; *Nordkvist, Jensen & Villadsen, 2003*). In particular, *Jensen et al. (2001)* reported that the levels of lactate and ethanol were almost similar at specific aeration level, but then decreased when aeration was increased. As shown in Fig. 2, the levels of lactate and ethanol were almost similar at 30WA. Meanwhile, formate was not detected in the extracellular of *L. lactis* under 30WA, 30WOA and 37WOA. It was demonstrated that reaction catalysed by PFL was inactivated under oxygen/ aerated conditions. Detection of different fermentation by-products at different growth conditions suggested a redirection of carbon flux in the pyruvate branch point. In particular, an increase in the flux of PDH may lead to reduction in NADH/NAD$^+$. Therefore alternative pathways are required to maintain redox balance of NADH/NAD$^+$ in *L. lactis*. On the contrary, ethanol was found as the major product when *L. lactis* was grown on maltose

**Table 2 Physiological parameters of *L. lactis* under different conditions.** Cultivation was carried out at 30 °C with agitation (150 rpm) (30WA), 30 °C without agitation (30WOA) and 37 °C without agitation (37WOA). The specific growth rate ($\mu$) (slope of growth curve) and biomass yield ($Y_{x/s}$) (coefficient linear regression ($R^2$)) obtained after five hours of cultivation.

| Parameter | 30 °C without agitation (30WOA) | 37 °C without agitation (37WOA) | 30 °C with agitation (30WA) |
|---|---|---|---|
| $\mu$ | 0.35 | 0.272 | 0.27 |
| $Y_{x/s}$ | 0.897 | 0.831 | 0.928 |

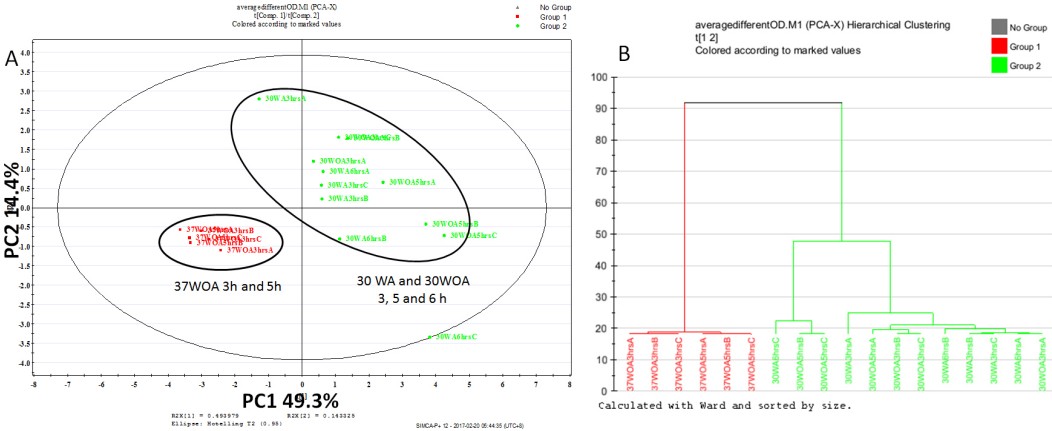

**Figure 3 Score plot of PCA (A) and dendrogram (B) of *L. lactis* at different conditions and different sampling points.** All conditions were obtained using 0.5 g/L of unlabeled glucose. Red represents 37 °C without agitation (37WOA), harvested at three and five hours, green represents 30 °C with (30WA) and without agitation (3OWOA) harvested at three, five and six hours of cultivation. Score plot was achieved via two principal components with total variation of 64%. Datasets of PAAs were normalized using D₄-alanine for normalization purposes.

(*Solem, Dehli & Jensen, 2013*). Finally, the physiological parameters (maximum specific growth rate $\mu$, biomass yield $Y_{x/s}$) highlighted in Table 2 were determined by regression analysis during the mid-exponential growth phase in the batch cultivations. It was found that 30WOA had the highest growth rate, whereas 30WA and 37WOA showed similar growth rates.

## The determination of sampling point

To assess the sensitivity of flux analysis to the sampling time point and different conditions, we compared the proteinogenic amino acid (PAA) profiles, obtained from three, five and six hours of cultivation. A total of nine biological replicates, represented 30WA, 30WA and 30WOA were initiated in parallel, harvested at different sampling times and evaluated using principal component analysis (PCA) (Fig. 3). The PCA score plot (Fig. 3A) indicated no discrimination patterns between different sampling time points for all conditions, suggesting that changes in metabolism are minimal when isotopic labeling of biomass was achieved throughout the cultivation. Specifically, the score plot showed that different sampling time points obtained from 37WOA were clustered together, whereas different sampling time points taken from 30WA and 30WOA were overlapping (Fig. 3B). In batch cultures, a metabolic pseudo-steady state is usually assumed during exponential growth phase (*Zamboni et al., 2009*). During this phase, cells are steadily dividing at their maximal

specific condition. It was suggested that labeling experiments should be continued long enough for metabolic pools to turn over several times to achieve isotopic steady state. For example, compared to glycolysis, the TCA cycle may require several hours to achieve isotopic steady state. Overall the PCA showed that cells can be harvested at five hours of cultivation with $OD_{600}$ between 0.8 and 1.2. Therefore, in later experiments, all cultures were harvested after five hours of incubation.

## Proteinogenic amino acid (PAA) profiles

Like other cells, environmental perturbations can have a significant impact on the metabolome of *L. lactis*. Amino acids are the basic substrates that act as regulators in many metabolic pathways. Thus amino acids can be viewed as a network that adapts to various physiological conditions. Consequently, detailed analysis of amino acids is important for $^{13}C$ metabolic flux ratio analysis because the analysis relies on labeling enrichment or mass distribution vectors (MDVs) which are independent of metabolite levels (*Buescher et al., 2015*). Using GC-MS, 16 PAAs (including ornithine) were detected in *L. lactis* after five hours of cultivation. As shown in Fig. 4, the concentration of PAAs is expressed as mM (mmol/L) as opposed to mM/gCDWh. The PAAs show clear variation in their concentration according to the respective conditions. Overall, the concentration of PAAs was higher in cells growing at 30WA than when growing at 30WOA and at 37WOA (Fig. 4A). Alanine and lysine were higher at 30WA and 30WOA, whereas aspartate, glutamate and glycine increased in 37WOA. Collectively, we compared the concentration of PAAs and intra- (Fig. 4B) and extracellular (Fig. 4C) free amino acids (FAAs). In addition to PAAs, determination of metabolic fluxes using FAAs has been described (*Mori et al., 2011*; *Okahashi et al., 2014*). PAAs are favourable for metabolic flux analysis because they are abundant in nature and easily extracted using hydrolysis. Interestingly, we found that the levels of intracellular FAA were higher than extracellular FAA and PAA levels. Specifically, the PAAs alanine, aspartate, glutamate and lysine were higher in all conditions, while intra- and extracellular alanine and glycine increased in all conditions. The abundance of these PAAs may influence the calculation of metabolic flux analysis and therefore should be taken into consideration.

Namely, variation in the levels of intra- and extracellular FAAs may indicate the specificity of cellular activities carried by *L. lactis*. Since *L. lactis* was cultivated in M17 medium, additional cofactors and nucleosides allow better transport of peptides than in defined media (*Adamberg, Seiman & Vilu, 2012*). Generally, the study of amino acid compositions in *L. lactis* under different environmental perturbations has been described (*Lahtvee et al., 2011*; *Marreddy et al., 2010*; *Papagianni, Avramidis & Filiousis, 2007*). Several amino acids are found in large amount including glutamate, serine, and aspartate. Namely, aspartate and glutamate are essential amino acids in *L. lactis* and are regarded as stimulating amino acids (*Adamberg et al., 2009*). These amino acids are sources of nitrogen, required during transamination to convert amino acids into the corresponding α-keto acids (*Pudlik & Lolkema, 2012*). Aspartate in particular is a precursor for the synthesis of five other amino acids, as well as for pyrimidine synthesis. Aspartate is also a nitrogen donor for purine biosynthesis and is further involved in synthesizing proteins, DNA, RNA, and ATP (*Wang et al., 2000*).

## Pearson correlation (*r*) analysis of PAAs

Typically, the selection of [13]C-labeled carbon sources depends on the metabolic flux distribution inside of the target cells. Therefore, different compositions of [13]C labeled and unlabeled glucose are often used to increase the precision of metabolic flux calculations and to resolve different pathways of central carbon metabolism. It was suggested that [1, 2-[13]C] glucose and a mixture of [1-[13]C] and [U-[13]C] glucose at 80%:20% are best to calculate flux levels of the pentose phosphate (PP) pathway, glycolysis and the TCA cycle, whereas a mixture of non-labeled, [1-[13]C], and [U-[13]C] glucose is suitable to measure the flux of the glyoxylate pathway reaction (*Fischer & Sauer, 2003*; *Fischer, Zamboni & Sauer, 2004*; *Maeda et al., 2016*). In particular, an investigation of useful carbon tracers for [13]C metabolic flux analysis in *L. lactis* has not been described. To investigate the correlation between PAAs in response to different compositions of [13]C labeled and unlabeled glucose and different conditions, we measured the level of PAAs and used Pearson correlation analysis to calculate the relationship between pairs of PAAs. Figure 5 shows two-way hierarchical clustering analysis heat map representation of the correlation matrices of PAA levels, obtained from 0%:80% (wt/wt) of [U-[13]C] glucose and unlabeled glucose (Fig. 5A) and 20%:80% (wt/wt) of [U-[13]C] glucose and [1-[13]C] glucose (Fig. 5B). The correlations were visualized in color-coded correlation matrices with color gradient ranging from red to green. In this representation, red indicates high positive correlation and green represents high negative correlation. In addition, we excluded alanine from the analysis because of inconsistent values found in labeling experiments.

The correlation matrices obtained from 20%:80% (wt/wt) of [U-[13]C] glucose and unlabeled glucose indicated that branched amino acids (valine, isoleucine, leucine) and aromatic amino acids (phenylalanine, tyrosine) were correlated strongly ($r \geq 0.7$) among PAAs in all conditions (Fig. 5A). Specifically, under 30WA and 30WOA, branched chain amino acids were positively correlated with proline and phenylalanine. Growth under 37WOA indicated that valine and leucine were positively correlated with proline. Meanwhile, correlation matrices obtained from 20%:80% (wt/wt) of [U-[13]C] glucose and [1-[13]C] glucose also revealed that correlations of branched amino acids (valine, isoleucine, leucine) were higher in all conditions (Fig. 5B). Specifically, leucine was positively correlated with serine, proline and threonine at 30WA. At 30WOA, glycine showed positive correlation with proline and aspartate. Threonine also showed strong correlation with branched chain amino acids at 30WOA, while the correlations between histidine and lysine were remarkably higher at 30WA and 37WOA (Fig. 5B). In particular, positive correlation with histidine was only observed when growing on the mixture of [U-[13]C] glucose and [1-[13]C] glucose.

Namely, the results suggest similar PAAs with strong correlation are found in all conditions, suggesting their consistent role in *L. lactis* in response to environmental perturbations. These PAAs can be regarded as the common amino acids found in *L. lactis* under different conditions, thus [13]C enrichment patterns of these amino acids are crucial for reliable metabolic flux analysis of *L. lactis*. Specifically, the interweavement of these amino acids indicates flux regulation of glycolysis, the pentose phosphate (PP) pathway and involvement of malate (MAL) and oxaloacetate (OAA). Namely, branched chain amino acids are associated with pyruvate (PYR), OAA and 2-oxoglutarate (OGA). Phenylalanine

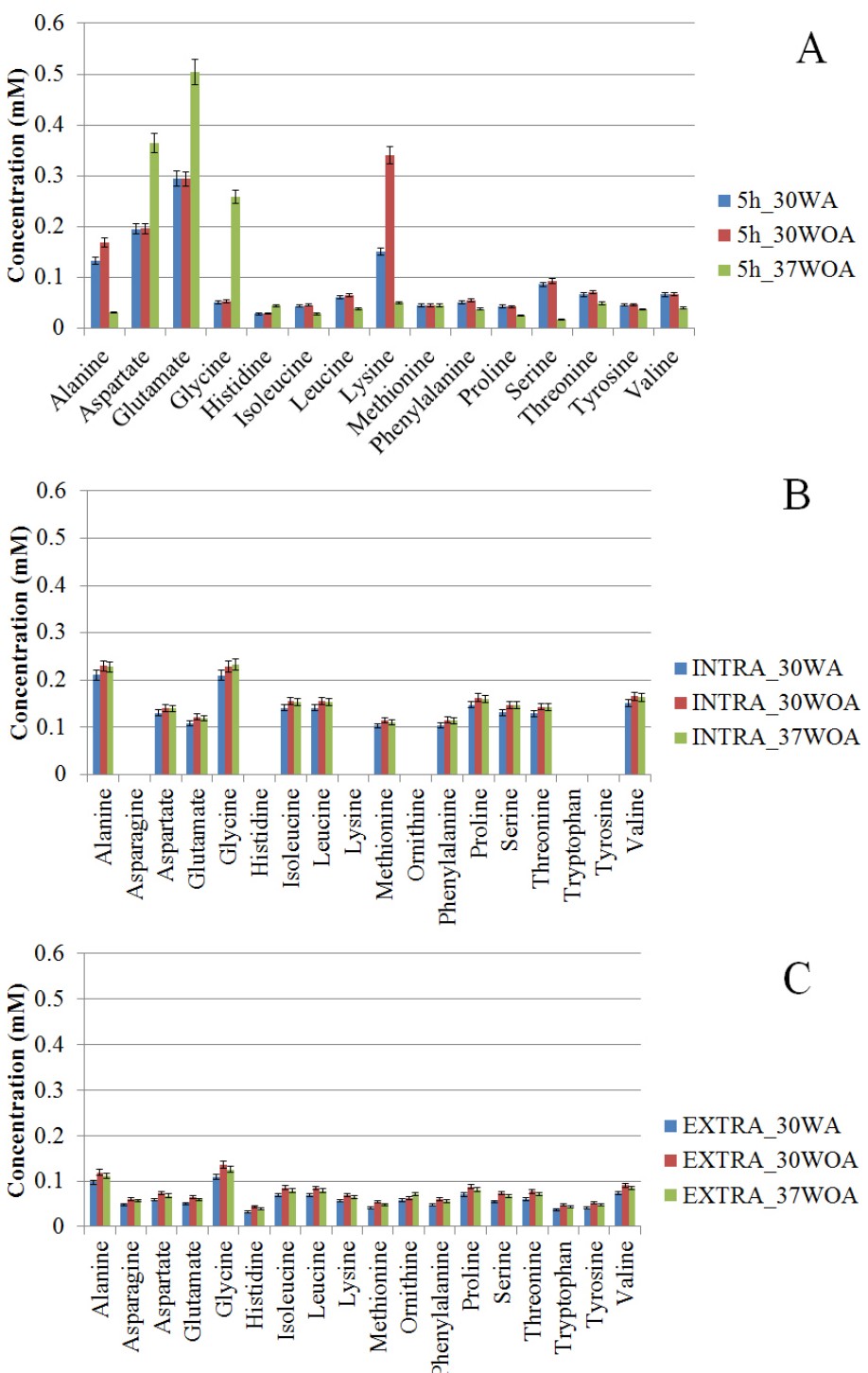

**Figure 4  Comparative analysis of amino acids at different concentrations (mM).** The proteinogenic amino acid (PAA) (A) and free amino acids (FAAs) obtained from intra- (B) and extracellular (C) of *L. lactis* were collected after five hours of cultivation. The concentrations of amino acid were obtained after five hours (5 h) of cultivation. 30WA represents 30 °C with agitation, 30WOA represents 30 °C without agitation and 37WOA represents 37 °C without agitation. Values are given as mean ± standard deviation ($n = 5$, biological replicates).

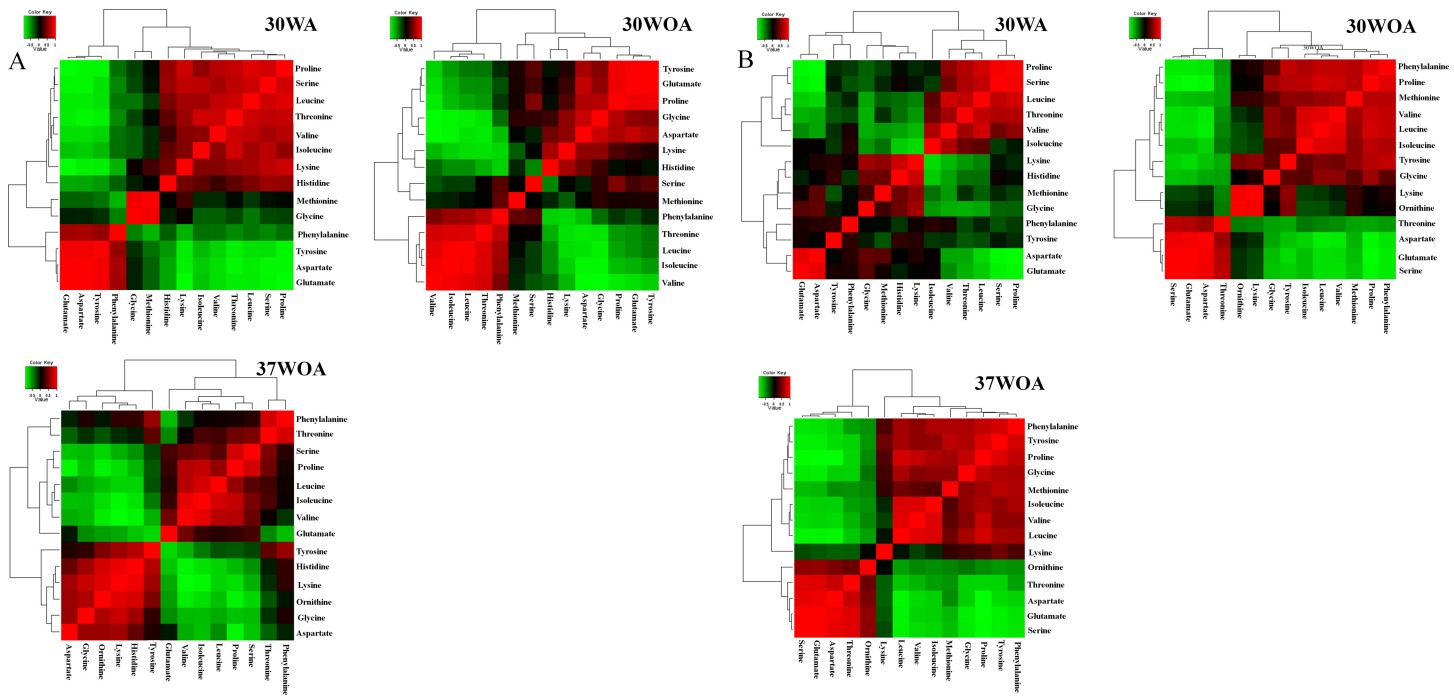

**Figure 5** **The correlation matrices of PAA profiles using Pearson correlation analysis ($r$).** *L. lactis* was grown on 20%/80% (wt/wt) of [U-$^{13}$C] glucose and unlabeled glucose (A) and 20%/80% (wt/wt) of [U-$^{13}$C] glucose and [1-$^{13}$C] glucose (B) at 30WA, 30WOA and 37WOA. Ornithine was included in the analysis. 30WA represents 30 °C with agitation, 30WA represents 30 °C with agitation, 30WOA represents 30 °C without agitation and 37WOA represents 37 °C without agitation.

and tyrosine are related to phosphoenolpyruvate (PEP) and erythrose 4-P (E4P). Lysine is contributed by acetyl-CoA (AcCoA) and OGA whereas histidine is linked to pentose-5-P (P5P). Overall, differences and similarities in the correlation patterns reveals that the PAAs can be interpreted as a fingerprint of the underlying cellular metabolism and could be useful for optimization of $^{13}$C labelling experiments in *L. lactis.*

## $^{13}$C Metabolic flux ratio analysis

To calculate the fractional contribution of central carbon metabolism, we used $^{13}$C metabolic flux ratio analysis that relies on the detection of PAAs labelling enrichment, also known as mass distribution vector (MDV) (*Buescher et al., 2015*; *Nanchen, Fuhrer & Sauer, 2007*; *Zamboni et al., 2009*). The MDVs of PAAs were used to calculate the fraction labeling (FL) of precursor intermediates and subsequently used to measure the flux ratios of active pathways in central carbon metabolism. Specifically, 24 FLs of precursor intermediates calculated from MDVs of PAAs were obtained (Fig. 6). Using these FLs, we then calculated the flux ratios (fraction of the total pool (%)) of active pathways and visualized the values according to the conditions (Figs. 7A, 7B). Each flux ratio map contains reactions of the glycolysis, the oxidative and non-oxidative pentose phosphate (PP) pathway, as well as metabolic node of oxaloacetate (OAA), catalyzed by pyruvate carboxylase (pycA), malic enzyme and serine and C1 metabolism and fermentation by-products (Fig. 7B) (*Flahaut et al., 2013*; *Hoefnagel et al., 2002*; *Jensen et al., 2002*). In particular, the size of stoichiometric

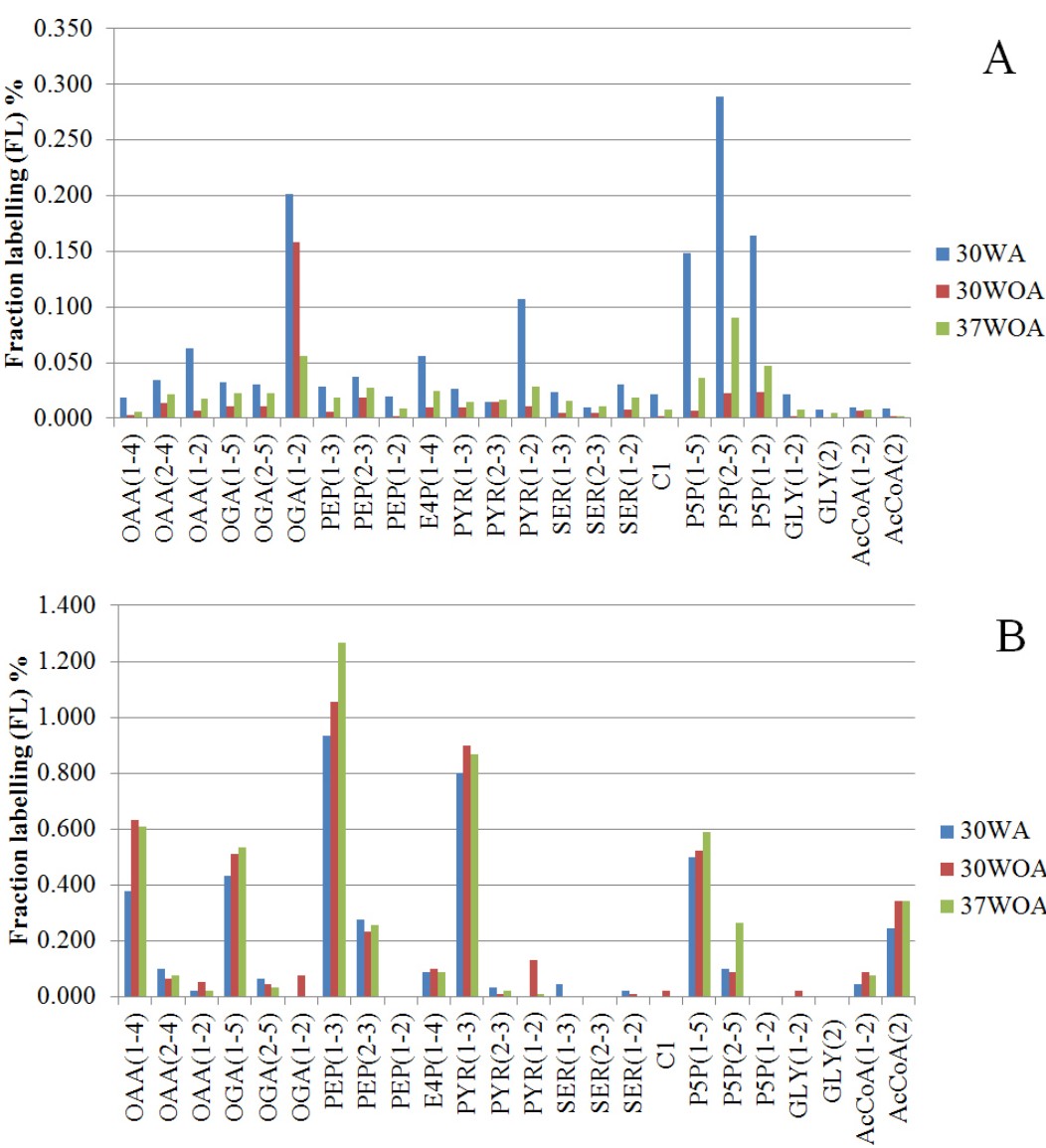

**Figure 6** **Comparisons of fraction labeling (FL) of precursor intermediates of central metabolism.** The data were calculated from MDVs of PAAs, obtained using different compositions of 20%:80% of [U-$^{13}$C] glucose and unlabeled glucose (A) and 20%:80% (wt/wt) of [U-$^{13}$C] glucose and [1-$^{13}$C] (B). 30WA represents 30 °C with agitation, 30WOA represents 30 °C without agitation and 37WOA represents 37 °C without agitation.

models for flux ratios is often reduced to lower the risk of numerical problems (*Kogadeeva & Zamboni, 2016*). In this study, model reduction was carried out by omitting several pathways and cofactors that have not been reported in *L. lactis*. In case of TCA cycle, *L. lactis* is reported to have an incomplete TCA cycle (*Vido et al., 2004*). Therefore the reaction is omitted except for malate (MAL) and oxaloacetate (OAA) which are involved in malic enzyme and reaction that creates OAA from PYR (*Wang et al., 2000*).

In the case for flux ratios for reactions catalysed by LDH, ACK and ADH, the carbon tracer (glucose) used in this study was not suitable to further resolve the fluxes. It was

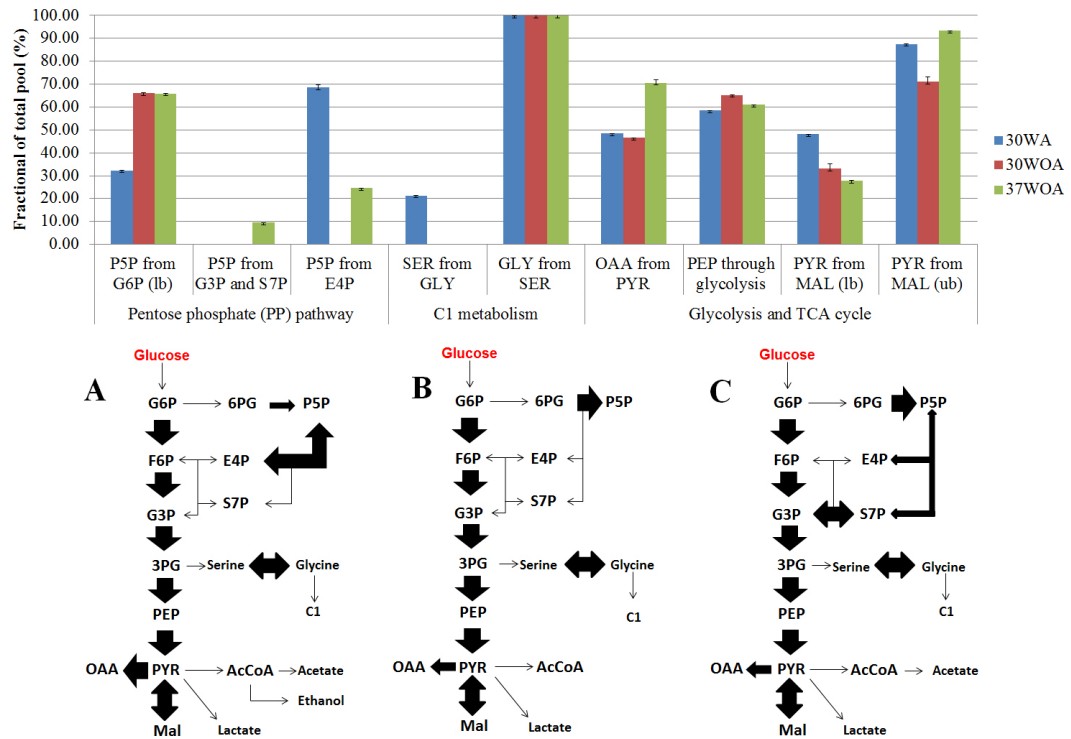

**Figure 7** **Simplified network models of *L. lactis* grown at different conditions.** (A) The flux ratios visualized in bar chart was calculated using Fiatflux. The experimental error (error bars) was estimated from redundant mass distributions using FiatFlux. (B) Simplified flux map that describes the central carbon metabolism of *L. lactis* when grown at 30 °C with agitation (30WA), 30 °C without agitation (30WOA) and 37 °C without agitation (37WOA). Each flux map includes the reactions of the glycolysis, the oxidative and non-oxidative pentose phosphate (PP) pathways, as well as the reactions catalyzed by pyruvate carboxylase, malic enzyme and serine and C1 metabolism. Fermentation by-products of lactate, acetate and ethanol were also included in the flux map. Abbreviation; G6P, glucose-6-phosphate; F6P, fructose-6phosphate; G3P, glyceraldehyde-3-phosphate; P5P, ribose-5-phosphate; S7P, sedoheptulose-7-phosphate; E4P, erythrose-4-phosphate; 3PG, 3-phosphoglyceric acid; PEP, phosphoenolpyruvate; pyruvate, PYR; oxaloacetate, OAA; malic acid, Mal; acetyl-CoA, AcCoA.

suggested that these pathway fluxes can be further elucidated using $[1, 2\text{-}^{13}C_2]$ glucose or fructose and/or by using a mixture of $[3\text{-}^{13}C]$ lactate and naturally labeled ethanol (*Adler et al., 2013*).

The 20%:80% of $[U\text{-}^{13}C]$ glucose and unlabeled glucose composition (Fig. 6A) indicated higher FLs of pentose-5-P (P5P) fragments at 30WA than other conditions. The results strongly suggest activity in the pentose phosphate (PP) pathway. Generally, the flux contributions of the pentose phosphate (PP) pathway can be measured via the transketolase (TK) and transaldolase (TA) reactions. TK catalyzes the transfer of a 2-C fragment containing a ketone group from xylulose-5-P (Xu5P) to the 5-C ribose-5-P (R5P) to produce sedudoheptulose-7-P (S7P) and G3P. TK may also transfer a 2-C fragment (containing a ketone group) from Xu5P to erythrose-4-P (E4P) to produce F6P and G3P. Meanwhile, TA facilitates the transfer of a 3-C from S7P to G3P to produce E4P and F6P. The fraction of the total pool (%) in Fig. 7A showed that the activity of TK and TA in the PP pathway was reflected by flux ratios of P5P from glucose-6-P (G6P), P5P from G3P

and S3P and P5P from E4P. P5P from glucose-6-P (G6P) was calculated as a lower bound reaction because TK can reversibly cleave P5P thus leading to multiple cycling through the PP pathway (*Fischer & Sauer, 2003*; *Sauer et al., 1999*). The flux ratios indicated 30% of P5P was generated from G6P at 30WA (Fig. 7A) and more than 65% of P5P was generated from G6P at 30WOA (Fig. 7A) and 37WOA (Fig. 7A). The fraction of P5P from G3P and S7P was 10% for 37WOA (Fig. 7A) whereas the fraction of P5P from E4P showed that 70% of P5P was generated from E4P at 30WA (Fig. 7A) and 25% of P5P was generated from E4P at 37WOA (Fig. 7A). The higher FLs of $P5P_{(2-5)}$ suggest that the C-3–C-4 carbon bonds in P5P were cleaved. The results also suggest that P5P was generated from fructose-6-P (F6P) and glyceraldehyde-3-P (G3P). The increase of flux in the PP pathway leads an increase of NADPH via the oxidative pathway. NADPH is required for many biosynthesis pathways and particularly important to reduce reactive oxygen species (ROS). As a facultative anaerobe, *L. lactis* very often encounter a challenge from oxygen. Therefore, *L. lactis* has to reorganize its metabolism in response to this challenge. Interestingly, flux ratios through the oxidative PP pathway increased at 30WOA (Fig. 7B) and at 37WOA only (Fig. 7A). Meanwhile, the non-oxidative phase is responsible for production of ribose-5-P for generating G6P and subsequently maintaining NADPH production.

The FLs of precursor intermediates obtained from 20%:80% (wt/wt) of [U-$^{13}$C] glucose and [1-$^{13}$C] (Fig. 6B) indicated higher FLs for fragments of oxaloacetate (OAA), OGA, PEP, pyruvate (PYR) and acetyl-CoA (AcCOA). FLs of PYR and OAA are useful to calculate the glycolysis and the PP pathway branch ratios. A combination of [U-$^{13}$C] glucose and [1-$^{13}$C] has generally been used to decipher the relative fluxes connecting the lower part of glycolysis and the TCA cycle. In this study, the flux ratios indicated that 60% of PEP was contributed by glycolysis, suggesting a substrate level phosphorylation in *L. lactis*, by converting sugars via the glycolytic pathway (*Neves et al., 2005*). Additionally, it has been reported that *L. lactis* growth on minimal glucose (55 mM) led to the formation of extremely homolactic fermentation by increasing the activities of phosphofructokinase (PFK), pyruvate kinase (PYK) and lactate LDH and therefore led to accumulation of lactate (*Papagianni, Avramidis & Filiousis, 2007*). As shown in Fig. 2, lactate is detected in all conditions, together with acetate and ethanol.

On the other hand, differences in the FLs of $PEP_{(2-3)}$ and $PYR_{(2-3)}$ suggest the activity of the malic enzyme reaction in *L. lactis*. Specifically the fraction of total pool (%) showed that the lower bound of PYR from MAL was below than 60% in all conditions and the upper bound of PYR from MAL almost exceeded 80% for all conditions. The fraction of PYR obtained from malate via malic enzyme were determined by comparing the MDV of $PYR_{(1-2)}$ and $MAL_{(1-2)}$. Additionally, this reaction contributes to the production of NADPH and therefore could play a role as alternative pathway to fulfil the requirement of NADPH in *L. lactis*. Moreover, regulation of malic enzyme could lead to different metabolic modes in response to environmental perturbation (*Adler et al., 2014*).

Higher flux ratios of OAA from PYR suggest the activation of pyruvate carboxylase (pycA) in *L. lactis*, which convert PEP or PYR respectively, to OAA. OAA is important in *L. lactis* as precursor for aspartate synthesis to synthesize other amino acids, as well as for pyrimidine synthesis and as nitrogen donor for purine biosynthesis (*Wang et al.,*

*2000*). At 30WA, 70% of OAA was contributed by PYR, whereas approximately 50% of OAA was contributed by PYR at 30WOA and at 37WOA. Activity of pycA suggests an intracellular acid tolerance response (ATR) in the *L. lactis* towards low pH. Changes in pH are contributed by accumulation of lactate and acetate, which were detected in the extracellular. Additionally, an increase of pycA suggests the conversion of PYR for regeneration of cofactors in glycolysis and the TCA cycle (*Budin-Verneuil et al., 2005*).

Meanwhile the reversible exchange of serine and glycine can be measured using the fraction of serine ($SER_{(1-3)}$) from glycine ($GLY_{(1-2)}$) and a C1 unit versus the fraction that is identical with $PEP_{(1-3)}$ (*Fischer & Sauer, 2003*). Lower FLs for serine (SER), glycine (GLY) and C1 suggest that serine and glycine were not interconverted and may not be used for C1 metabolism. However, approximately 20% of flux ratios of SER from GLY were calculated at 30WA. In particular, the reaction was reported to enhance the production of NADPH (*Fan et al., 2014*). Since the production of NADPH through the oxidative PP pathway at 30WA was lower due to lower flux, this reaction could become an alternative pathway to facilitate the NADPH requirement under the particular condition. Additionally, it was suggested that *L. lactis* has developed other pathway to provide NADPH as *L. lactis* is known to have incomplete TCA cycle, which is used to recycle the NADH pool for the respiratory chain (*Vido et al., 2004*).

## CONCLUSIONS

In this study, we used $^{13}$C-based metabolic flux analysis and GC-MS to determine the fractional contributions of *L. lactis* central metabolism in response to 30WA, 30WOA and 37WOA. By using different compositions of $^{13}$C labeled and unlabeled glucose, the variability and pairwise correlation of PAAs and the flux ratios of active pathways were measured. Intracellular FAAs were higher than extracellular FAAs and PAAs. PAA correlation analysis suggested that several amino acids were strongly correlated in all conditions. Branched chain and aromatic amino acids, alanine, aspartate, glutamate and glycine were identified as the common amino acids observed in *L. lactis* under different conditions and can be regarded important for reliable $^{13}$C based metabolic flux analysis for *L. lactis*. Namely, glucose was mainly converted via glycolysis in *L. lactis*, despite the different conditions. The flux ratios for the oxidative phase PP pathway were relatively high at 30WOA and 37WOA, indicating the NADPH requirement in response to oxygen related stress. Malic enzyme activity was also detected in *L. lactis* based on differences found in the FLs of PEP and PYR. Higher flux ratios through OAA from PYR suggested intracellular acid tolerance response (ATR) in response to pH stress in *L. lactis*. Taken together, our results provide insights into the cellular physiological state of *L. lactis* and a systems-level understanding of how *L. lactis* regulates central metabolism in various conditions.

### Funding
This work was supported by the Ministry of Science, Technology and Innovation Malaysia (MOSTI) (02-01-02-SF0987 and DLP-2013-024). The funders had no role in study design, data collection and analysis, decision to publish, or preparation of the manuscript.

### Grant Disclosures
The following grant information was disclosed by the authors:
Ministry of Science, Technology and Innovation Malaysia (MOSTI): 02-01-02-SF0987, DLP-2013-024.

### Competing Interests
The authors declare there are no competing interests.

### Author Contributions
- Kamalrul Azlan Azizan performed the experiments, analyzed the data, wrote the paper, prepared figures and/or tables, reviewed drafts of the paper.
- Habtom W. Ressom and Eduardo R. Mendoza wrote the paper, reviewed drafts of the paper.
- Syarul Nataqain Baharum conceived and designed the experiments, contributed reagents/materials/analysis tools, wrote the paper, reviewed drafts of the paper.

### Data Availability
The raw data has been supplied as Supplemental files.

### Supplemental Information
Supplemental information for this article can be found online at http://dx.doi.org/10.7717/peerj.3451#supplemental-information.

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
