# Peer review of "C based proteinogenic amino acid (PAA) and metabolic flux ratio analysis of Lactococcus lactis reveals changes in pentose phosphate (PP) pathway in response to agitation and temperature related stresses"

_PeerJ, doi:10.7717/peerj.3451_

## Round 0.1 · original submission · Major Revisions

Our reviewers have identified some points of concern, the most salient of which is the choice of a late-exponential sampling point instead of points that are clearly on the exponential phase. Please address also all the other issues concerning data presentation and analysis.

Reviewer 1 ·

Basic reporting

This research article presents changes of fluxes in the oxidative pentose phosphate pathway under different growth conditions. I have some questions for some of the figures:
1. In figure 2, you presented the intracellular amino acid concentration as ‘relative’ in ‘ppm’. For steady-state flux analysis, one does not need to evaluate the free amino acid concentrations. What is the purpose of showing this figure? What concentration are those values relative to? In addition, most papers show the concentration in ‘mmol/L’. You may want to verify the unit.
2. In Figure 4, I would suggest you improve the flux map, for example, you can use different line thicknesses to present fluxes of different values.

Experimental design

1. It would be better to collect samples at two time points during the exponential growth phase. However, you seemed to have collected one sample at the late exponential growth phase under each condition. Can you justify this experimental design?
2, 0.5g/L glucose is be very low. Why did you use this concentration? Is this concentration commonly used for cultivating L. lactis?

Validity of the findings

1. The authors need to clarify which software or algorithm they used for correcting mass isotopomer distributions.
2. Before carrying out flux analysis, you need to confirm which pathways are present and which are not present in the central metabolism. You can use gene annotations from any database.
3. In the supporting file, you need to provide all the reactions you used for the flux analysis model. Specifically, did you measure the biomass compositions of L. lactis to obtain a precise biomass formation equation?
4. Did you measure the lactate concentrations? Did the strain have other byproducts during growth? These information should be considered in flux analysis.

Additional comments

In general, I think there are many things that needs improving before publication. Besides those mentioned above, I also have the following concerns:
1. What is the purpose of showing Pearson correlations among amino acids? You have described the results, but did not give clear explanation for this analysis. Why did you choose three different combinations of 13C-tracers?
2. FiatFlux is fully capable of performing flux analysis for the central metabolism. Why did you get only flux ratio analysis instead of a complete flux map? If you obtained a complete flux analysis, you can gain more information of this bacteria.

Reviewer 2 ·

Basic reporting

1. Introduciton is too long with wrong structure.
This study is an application of the establoished methodology to the Lactococcus lactis analysis.The manuscript should be started from
the section on L. lactis lines from 82 to 93. First and second section in the introduction (lines 46-69) must be reduced to several lines.

Experimental design

2. Wrong experimental design. In order to analyze pseudo steady state by batch culture, cells have to be collected from the culture under a exponential growth phase (approximately 3-4 h in Fig 1.). The sampling time employed in this study is too late to consider pseudo steady state.
Specific growth rate, specific glucose consumption rate, and lactate production rate are at least required for the MFA study of microbe. Please analyze amino acid levels in culture broth in time dependent manner to investigate the amino acid uptake.

Validity of the findings

3. Fig. 2 Please do not use plan excel graphs. There is no information on 3h and 5h in the figure, as well as the experimental replicates in the legend.. What is the relative concentration with unit ppm ???

4. lines 216-228. What is [1] 13C glucose?. Please follow IUPAC rules.

5. Fig. 3 The figure has no information on the metabolic flux. Please remove from the manuscript. Instead of Fig. 3, raw MVD data for the key amino acids to estimate the flux ratio should be showen and explained in the manuscript.


6. Fig. 4. I cannot understand the value in the figure. No G6P was used for the cell growth? 37WOA-65.86% means that the flux ratio was estimated within 65.85-65,87%? How the super precise analysis was achieved? There is no legend on the bar graph. The bar graph as no legend on Y axis. What do error bars represent in the bar graph?
What is the bold line in part of the Glyoxylate shut reaction?.

7. Please show the MVA data representing the large the OAA from GOX flux level in 37 WOA to confirmed the result.

·

Basic reporting

The authors studied the flux changes in response to agitation and temperature related stresses in Lactococcus lactis. Using metabolic flux ratio (METAFoR) analysis, the changes in pentose phosphate (PP) pathway has been identified. Experiments were conducted in a rational manner, and conclusion based on the results was reasonable. However, some minor issues should be addressed to improve the manuscript quality.
(1) Redraw figure 2, the pattern and color of the bar chart is difficult to see clearly.
(2) Please combine Figure 4A and 4B and rearrange the pathway and the bar chart.

Experimental design

The authors conducted this study in three conditions 30WA, 30WOA and 37WOA. All the procedure is well designed and the analysis software is decent and well justified. The only question is that why not put 37WA into consideration?

Validity of the findings

As Lactococcus lactis subsp. cremoris MG1363 is an important starter culture for dairy fermentations. It is important to understand the basics of the metabolic patterns under different fermentation conditions. This paper first use the 13C technology to reveal the metabolic flux under different flux. The Data is robust and statistically controlled. The paper is elucidative on the data generated from experiment. But the biological mechanism behind the data well has not been fully discussed and explained. More discussion should be added to the relationship between the stress and the corresponding changes of the flux.

---

## Round 0.2 · Minor Revisions

In my opinion, you have adequately addressed most issues raised by our reviewers. I still have a few requests additional requests:
- line 345 states that TCA reactions were not included, whereas line 181 states that TCA reactions were included in the model. Please clarify
- An analysis of the resons behind the different amouns of lactate, ethanol and acetate in the three growth conditions is missing. This should be discussed either through comparison with relevant literature or with the outcomes of your flux analysis. Please state also why the relevant reactions (those catalyzed by lactate dehydrogenase, alcohol dehydrogenase, pyruvate-formate lyase and acetate kinase) were not included in the model.
- The font size in several of your figures is too small. Please adjust.

·

Basic reporting

The authors studied the flux changes in response to agitation and temperature related stresses in Lactococcus lactis. Using metabolic flux ratio (METAFoR) analysis, the changes in pentose phosphate (PP) pathway has been identified. Experiments were conducted in a rational manner, and conclusion based on the results was reasonable.
The figures were improved based on the comments and could meet the requirements for publication. The discussion regarding changes in PAAs, fraction labelling and flux ratios have been added accordingly to explain the biological mechanism behind the data.
The version meets the requirements for the publication.

Experimental design

Experiments were further improved and conclusion based on the results was reasonable.

Validity of the findings

No comments

---

## Round 0.3 · accepted · Accept

I am satisfied with your latest additions.